# Dexamethasone Downregulates Autophagy through Accelerated Turn-Over of the Ulk-1 Complex in a Trabecular Meshwork Cells Strain: Insights on Steroid-Induced Glaucoma Pathogenesis

**DOI:** 10.3390/ijms22115891

**Published:** 2021-05-31

**Authors:** Diego Sbardella, Grazia Raffaella Tundo, Massimo Coletta, Gianluca Manni, Francesco Oddone

**Affiliations:** 1IRCCS-Fondazione Bietti, 00198 Rome, Italy; graziaraffaella.tundo@fondazionebietti.it; 2Department of Clinical Sciences and Translational Medicine, University of Tor Vergata, 00133 Rome, Italy; coletta@seneca.uniroma2.it (M.C.); gianlucamanni53@yahoo.com (G.M.)

**Keywords:** autophagy, ubiquitin proteasome system, intraocular pressure, glaucoma, glucocorticoids, trabecular meshwork

## Abstract

Steroid-induced glaucoma is a severe pathological condition, sustained by a rapidly progressive increase in intraocular pressure (IOP), which is diagnosed in a subset of subjects who adhere to a glucocorticoid (GC)-based therapy. Molecular and clinical studies suggest that either natural or synthetic GCs induce a severe metabolic dysregulation of Trabecular Meshwork Cells (TMCs), an endothelial-derived histotype with phagocytic and secretive functions which lay at the iridocorneal angle in the anterior segment of the eye. Since TMCs physiologically regulate the composition and architecture of trabecular meshwork (TM), which is the main outflow pathway of aqueous humor, a fluid which shapes the eye globe and nourishes the lining cell types, GCs are supposed to trigger a pathological remodeling of the TM, inducing an IOP increase and retina mechanical compression. The metabolic dysregulation of TMCs induced by GCs exposure has never been characterized at the molecular detail. Herein, we report that, upon dexamethasone exposure, a TMCs strain develops a marked inhibition of the autophagosome biogenesis pathway through an enhanced turnover of two members of the Ulk-1 complex, the main platform for autophagy induction, through the Ubiquitin Proteasome System (UPS).

## 1. Introduction

The term glaucoma identifies a heterogeneous group of neurodegenerative disorders characterized by a loss of retinal ganglion cells (RGCs), degeneration of the optic nerve and irreversible blindness [1,2,3].

Glaucoma is a multifactorial disease whose prevalent clinical presentation is represented by primary open-angle glaucoma (POAG), generally correlated with ageing, chronic redox imbalance and increased ocular pressure (IOP) [4,5,6].

IOP increase is supposed to be caused by the impaired drainage of the aqueous humor, a fluid which shapes and nourishes the tissue of the anterior segment of the eye through anatomical outflow pathways. The main outflow pathway is the trabecular meshwork (TM), a specialized tissue located at the iridocorneal angle through which the aqueous humor is drained into episcleral veins [3,7]. The TM is synthesized by several resident cells and, mostly, by Trabecular Meshwork Cells (TMCs), an endothelial-like histotype with phagocytic and secretive properties [2,7,8,9].

A progressive IOP increase is observed in steroid-induced glaucoma, a severe clinical form of the disease, which is diagnosed in a subset of subjects who adhere to glucocorticoids (GC) therapy [10,11,12]. Steroid-induced glaucoma commonly affects young individuals, runs asymptomatically until the late stage, and may easily lead to impaired visual function and irreversible blindness before being diagnosed [12].

The pathogenesis of steroid-induced glaucoma has been long linked to the excessive secretion and TM deposition of myocilin (MYOC/TIGR gene), an intracellular and extracellular glycoprotein with functions that are still unclear. Since TMC is the only cell type in the organism synthesizing and secreting myocilin as a secondary GC response, steroid-induced glaucoma pathogenesis has been commonly viewed to rely on a dysregulation of the metabolism of these cells [8,13,14]. However, recent findings in cell-based models and in transgenic mice have questioned the relevance of myocilin for disease onset [15,16].

On the other hand, some independent research has suggested that regulation of the intracellular proteolytic pathways, namely the Ubiquitin Proteasome System (UPS) and autophagy, in TMCs, might be deeply involved in glaucoma onset and progression [16,17,18,19].

UPS and autophagy display a sophisticated and hierarchical organization which has evolved through the acquisition of non-redundant and complementary roles in surveying intracellular protein homeostasis, also referred to as proteostasis [20]. The UPS carries out the clearance of selected intracellular substrates, mostly short-lived proteins. Substrates are enzymatically conjugated with ubiquitin (Ub) moieties (through the E1–E2–E3 enzymes cascade) and proteolytically digested by the multi-subunit complex called proteasome [21,22,23]. The canonical proteasome is made up by (ia a hollow barrel-shaped catalytic core, called 20S, composed of four heptameric rings, two outer α rings and two inner β rings, which house the catalytic subunits encoding for the chymotrypsin-like, trypsin-like and caspase-like activities, and (b) one or two Regulatory Particles (RPs), called 19S, which couple the ATP-hydrolysis with recognition, unfolding and translocation of the poly-Ub substrates into the 20S [21,22,23].

Interestingly, in living cells, the proteasome exists in different structural configurations, referred to as doubly capped (19S–20S–19S), single-capped (19S–20S) and free 20S proteasome [22]. Although the substrate specificities of each complex are still unclear, the 20S is supposed to deal with clearance of oxidized, natively unfolded and misfolded proteins, while capped assemblies are involved in the clearance of poly-ubiquitinated proteins [23,24,25,26]; their relative distribution among various assemblies in the intracellular microenvironment is then tuned depending on the cell’s metabolic needs.

Unlike UPS, the autophagy pathway, and specifically macroautophagy (i.e., the most studied one, since microautophagy and chaperone-mediated autophagy are not discussed herein), proceeds through the formation of autophagosomes (i.e., double-membrane vesicles) to sequester a discrete portion of cell cytosol and organelles. These cargos are then delivered to lysosomes for clearance by hydrolytic enzymes [27,28,29]. Through this “self-eating process”, the cell is able to recycle nutrients from unnecessary or damaged proteins/organelles under starving conditions, although it is now clear that almost all cells preserve their energy metabolism through a low-rate basal autophagy even under nutrient-rich conditions [30].

Autophagosomes biogenesis and “flux” toward lysosomes is governed by an integrated signaling network, which encompasses a plethora of enzymes and kinases (e.g., AMP-activate Kinase (AMPK), mammalian target of rapamycin (mTOR)/Akt, among others), sensing the metabolic and energetic status of the cell and transducing the input toward the activation or repression of autophagy through the recruitment of a myriad of Autophagy-Related Genes (Atg) [27,30,31,32,33]. A main molecular platform which acts as a node for these signaling pathways is the Unc-51-like Autophagy Activating Kinase 1 (Ulk-1) complex [32,34]. This macromolecular complex is composed of Ulk-1 or Ulk-2, FAK family kinase-interacting protein of 200 kDa (Fip200, also called RB1CC1), Atg13 and Atg101 and it is supposed to be stable in the cell cytosol, even under nutrient-rich conditions [34].

Upon activation by upstream effectors, the Ulk-1 complex phosphorylates downstream Atg(s), stimulating the nucleation and assembly of the autophagosome membrane through the recruitment, at initiation sites, of main effectors, namely the class III phosphatidylinositol 3-kinase (PtdIns3K) complex, which is composed of the PtdIns3K Vps34 (vacuolar protein sorting 34), Vps15 (p150 in mammalian cells), Atg14 and Beclin 1 [27,34]. Thereafter, the PtdIns3K complex releases PtdIns3P (phosphatidylinositol 3-phosphate), which recruits PtdIns3P-binding proteins (such as Atg18, Atg20, Atg21, and Atg24) to the nascent membrane of autophagosome (e.g., phagophore) together with two interrelated ubiquitin-like (Ubl) conjugation pathways, namely (i) Atg12–Atg5–Atg16, mediated by Atg7, and (ii) phosphatidylethanolamine (PE)-conjugated Atg8, also known as microtubule-associated proteins 1A/1B (LC3B) (which is the most studied autophagy marker to analyze autophagy flux, especially in vitro) [28,35,36]. These Atg(s) then carry out the elongation and fusion of the autophagosome vesicle, generating a mature double-membrane autophagosome [27,35,36].

Although autophagy was initially supposed to carry out a bulky non-selective clearance of substrates, sophisticated mechanisms of cargo sorting have been unveiled. In this regard, it is worth mentioning the role of p62-sequestosome (p62/SQSTM1) in shuttling ubiquitinated proteins inside the growing autophagosomes [35,36,37].

In this framework, several stimuli, in particular when chronically administered, turn into a complex dysregulation of autophagy and/or UPS by virtue of the tight interplay between these pathways, cell metabolism and growth, proliferation and apoptosis [33,38].

On the basis of these considerations, there is the possibility that GCs treatment might induce an alteration of intracellular proteolytic pathways in primary TMCs

The profound effects on cell metabolism that GCs are known to induce in vitro and in vivo elicited an investigation into the regulation of UPS and autophagy in a cultured human TMC strain stimulated with dexamethasone (hereafter referred to as Dexa), a prototypical GC, by following different dosage and timing of delivery. This made it possible to cast light on a UPS-related severe impairment of autophagosome biogenesis, especially in TMC cells repeatedly stimulated with Dexa over 6 days, hinting a possible molecular pathway for interpreting the pathogenesis of steroid-induced glaucoma.

## 2. Results

### 2.1. Prolonged TMCs Stimulation by Dexa Reduces Cell Growth and Induces Apoptosis

To validate cell identity, myocilin expression was first monitored in TMCs cultivated over 2 days (hereafter referred to as Dexa-2d) and over 6 days (hereafter referred to as Dexa-6d) in a medium supplemented with 1 µM Dexa; as the internal control, cells were treated with the Dexa solvent (ethanol) (hereafter referred to as vehicle cells). Thereafter, myocilin content was assayed by Western blotting (Wb) and RT-PCR. In accord with literature data [39], myocilin immunostaining significantly increased in whole cell lysates of Dexa-2d cells, but not in vehicle cells (Appendix A, *p* < 0.001), displaying an about 10-fold enrichment of myocilin in Dexa-6d cells with respect to vehicle cells (Appendix A, *p* < 0.00001). RT-PCR data confirmed that the transcription of myocilin mRNA was largely increased in Dexa-2d cells (i.e., an average 3-fold increase with respect to vehicle cells, *p* < 0.0001) and persistently elevated in Dexa-6d cells (i.e., an average 2.5-fold increase, *p* < 0.0015) (Appendix A).

Once the identity of cultured TMCs was verified, the proliferation and viability of Dexa-2d and Dexa-6d cells, as determined by MTT assay, was compared to that of vehicle cells at each timepoint (Figure 1A). However, an additional experimental condition was included in the study, namely the daily administration of 0.1 µM fresh Dexa for a total of 6 days (this experimental condition is hereafter referred to as Dexa-6d*).

The MTT approach indicated that the proliferation/viability of cells was unaffected by Dexa treatment for 2 days (Dexa-2d), whereas Dexa-6d showed an average 15% decrease in cell count (*p* < 0.005); Dexa-6d* cells displayed, instead, an average 35% reduction in cell count (*p* < 0.005). A qualitatively similar result was obtained by Trypan-blue cell count analysis under a light microscope equipped with a cell count software (Appendix A).

This effect stimulated a further semiquantitative analysis on apoptosis markers and protein regulating cell cycle progression by Wb. With respect to the corresponding vehicle-treated cells, the immunostaining of PARP p25 fragment, a recognized marker of apoptosis, was unaltered in Dexa-2d cells, slightly increased in Dexa-6d cells (*p* < 0.04) (Figure 1A,B) and robustly increased in Dexa-6d* cells (*p* < 0.0008) (Figure 1B).

Since these findings suggest a reduced proliferation and increased apoptotic frequency, Dexa-2d, Dexa-6d and Dexa-6d* cells were further probed for p21 and p53 proteins content. Interestingly, both p21 and p53 immunostaining displayed a significant increase in Dexa-2d cells (with respect to vehicle-2d cells) (*p* < 0.0001) (Figure 1C), while in Dexa-6d and Dexa-6d* cells, the p21 content turned back to the level of vehicle-treated cells (Figure 1C).

On the other hand, since p53 was persistently increased, phosphorylation of this protein at serine 46 (p53(Ser46)), which often labels its enrolment in the apoptosis cascade, was assayed (Figure 1C). Phospho-p53(Ser46) immunostaining was not significantly altered in Dexa-2d vs. vehicle-2d and in Dexa-6d vs. vehicle-6d cells, whilst a 2-fold increase was detected in Dexa-6d* vs. vehicle-6d* cells (*p* < 0.0001), indicating that the phospho-p53/p53 ratio was enhanced under this last experimental condition (Figure 1C). However, this ratio was lower in Dexa-2d vs. vehicle-2d cells (*p* < 0.008) and comparable between Dexa-6d vs. vehicle-6d cells (Figure 1C), indeed suggesting that only a continuous supply of fresh Dexa for 6 days significantly induced cell apoptosis through the p53 pathway in TMCs.

### 2.2. Proteasome Activity Is Not Affected by Dexa Treatment

The increased frequency of apoptotic cells in Dexa-treated cells, and in particular, in Dexa-6d* subgroup, raised the question on the functionality of the intracellular proteolytic pathways which play a key role in modulating cell metabolism and viability.

To first address this point, bulk proteasome activity of vehicle and Dexa-treated cells was tested by the so-called “proteasome assay” in crude cell extracts (i.e., the soluble fraction of cytosol isolated under denaturing conditions). Briefly, 20 µg of cells extracts were probed for the kinetics of LLVY-amc hydrolysis by the chymotrypsin-like activity of proteasome. The reaction was followed over 2 h and the fluorescence release was monitored until linearity was observed. Bulk proteasome activity was unaffected by Dexa treatment over the whole scheme and dosage of administration, with the exception of a slight increase in the case of Dexa-2d cells vs. vehicle-2d cells, even though its statistical significance was limited (Figure 2A).

To validate this finding, the same crude cell extracts were further analyzed by native gel electrophoresis. In this way, the main proteasome assemblies, commonly found into the cell cytosol (i.e., double-capped, single-capped and free 20S), were separated by mass/charge and probed with 75 µM LLVY-amc in-gel (Appendix A) [40]. It must be outlined that, since data from vehicle-treated cells at different days were fully comparable, for the sake of clarity, only one vehicle lane is reported.

The rate of peptide hydrolysis, which is linearly correlated to the light intensity recorded, displayed no major differences among the experimental groups. However, a very small decrease in capped particles activity, mirrored by a slight increase in that of free 20S, was documented in the case of Dexa-6d* cells (Appendix A).

To further verify their identity, the particles were probed with an anti-α7 antibody (i.e., a 20S subunit present in all assemblies) upon transfer to a nitrocellulose filter (Appendix A). This analysis showed a slight increase in the capped assemblies content in Dexa-2d cells with respect to vehicle cells; conversely, a less intense immunostaining of the same assemblies was detected in Dexa-6d* cells, but mirrored by an increase in that of free 20S (see capped assemblies/20S ratio quantification) (*p* < 0.003) (Appendix A). No significant differences were observed for Dexa-6d cells with respect to vehicle cells (Appendix A).

Since the structural composition of proteasome particles can be quickly modulated by external and internal stimuli, a possible explanation of the small increase in capped assemblies in Dexa-2d cells might derive from a structural rearrangement of the proteasome population soon after the Dexa treatment, which then fades out after 2 days of treatment [41]. With this in mind, the modulation of proteasome assemblies composition by Dexa stimulation at two different concentrations (i.e., 0.1 µM and 1 µM) was more deeply investigated after 1 day (referred as Dexa-1d). The native gel study showed after 1 day of stimulation a slight increase in the activity of capped assemblies (as from LLVY-amc staining), backed by a marked increase in their content (as from α7-antibody immunodetection) in Dexa-treated cells (*p* < 0.0004 and *p* < 0.0001 for 0.1 µM and 1 µM Dexa, respectively) (Appendix A). Administration of either 0.1 µM or 1 µM Dexa for 2 days displayed a fully overlapping effect with that described in Appendix A and is not further commented on.

A further semiquantitative analysis of α7 and Rpn10 subunits (i.e., a 19S subunit which is supposed to correlate with the stability of the 19S mature particle) was carried out by Wb in whole cell lysates from each experimental group (ranging from Dexa-1d to Dexa-6d), but no difference was detected under all tested experimental conditions (Figure 2B).

Thereafter, a semiquantitative analysis of poly-ubiquitinated proteins (i.e., the natural substrates of proteasome) in whole lysates was undertaken by Wb (Figure 2C). As compared to vehicle cells, Dexa-1d cells did not display any difference, whilst Dexa-2d showed a modest but significant decrease in poly-Ub proteins content (*p* < 0.04), which was even greater in Dexa-6d cells (*p* < 0.005) and definitively very marked in Dexa-6d* cells (*p* < 0.0001) (Figure 2C). In all cases, no obvious changes in the abundance of free Ub chains nor in that of free Ub monomer were observed (Figure 2C).

### 2.3. Autophagy Flux Is Impaired in Dexa-Treated Cells

The lack of any significant impairment of proteasome functionality following Dexa treatment stimulated an investigation on the autophagy flux under the same experimental conditions.

The most used autophagy marker is LC3B-II and the semiquantitative analysis of LC3B-II accumulation in the presence of an inhibitor of lysosomes hydrolases or autophagosomes/lysosomes fusion provides a reliable read-out of the overall autophagy flux [28,36].

Thus, vehicle cells along with Dexa-2d, Dexa-6d and Dexa-6d* cells were stimulated with 20 µM chloroquine (CQ) over the last 2h of treatment. Thereafter, whole cell lysates were analyzed by Wb and probed with an anti-LC3B antibody raised against the N-terminus of the protein. The addition of CQ brought about a consistent accumulation of LC3B-II in the case of vehicle cells (*p* < 0.0001) (Figure 3A), while Dexa-2d and Dexa-6d cells displayed only a modest increase in LC3B-II in the presence of CQ (*p* < 0.0001), much lower than that observed for vehicle cells; this gap was particularly evident for Dexa-6d cells (*p* < 0.0001) (Figure 3A). Finally, in the case of Dexa-6d* cells, the increase in LC3B-II in the presence of CQ was nearly undetectable (Figure 3A).

In order to confirm the progressive decrease in autophagy flux, two additional autophagy markers (i.e., p62/SQSTM1 and Beclin-1) were then assayed by Wb (Figure 3B) As a matter of fact, in Dexa-6d* cells, a tendency toward an increase in p62/SQSTM1 immunostaining was observed, a finding compatible with a defective autophagy (Figure 3B). Since, p62/SQSTM1 is supposed to be digested by lysosomal hydrolases, the quantification of this protein was further carried out in the presence of CQ in Vehicle-6d* and Dexa-6d* cells. As expected, in the presence of CQ, p62/SQSTM1 immunostaining turned out to increase in vehicle cells (*p* < 0.005), but not in Dexa-6d* cells (Figure 3C).

Likewise, basal Beclin-1 immunostaining displayed a marked tendency to decrease in Dexa-treated cells, and maximally in Dexa-6d* cells, even though no statistical significance among different independent experiments was reached. In this regard, it should be recalled that p62/SQSTM1 and Beclin-1 quantification must always be interpreted with caution as these markers do not unequivocally reflect the overall functioning of autophagy and they can be cleared out by intracellular caspases during apoptosis induction [42,43].

Therefore, in order to find further convincing evidence about a defective autophagy in Dexa-treated TMCs an immunofluorescence (IF) microscopy study was carried out on Dexa-6d* cells. The study was limited to this experimental group since IF studies are not very sensitive and Dexa-6d* cells were those displaying the apparent highest degree of autophagy inhibition.

Vehicle cells and Dexa-6d* cells were probed with an anti-LC3B antibody and observed under an immunofluorescence microscope at 40× and 100× magnification. It appears immediately evident that autophagosomes (i.e., the red dotted vesicles) were much less frequent in Dexa-6d* cells than in vehicle cells (Figure 4). Interestingly, Dexa-treated cells were characterized by the accumulation of very large vacuolar-like structures with a positive staining for LC3B.

Autophagosomes were then quantified either by means of cells displaying at least 30 LC3B^+^ dotted structure (*p* < 0.0001), a threshold selected on the basis of preliminary settings, and/or by the number of autophagosomes per cell (*p* < 0.004) (Figure 4). In both cases, a very significant impairment in autophagosomes biogenesis emerged in Dexa-6d* cells, confirming the autophagy dysfunction.

### 2.4. Downregulation of the ULK1 Complex in Dexa-Treated Cells

Since the IF study suggested that the defective autophagy observed in Dexa-6d* cells might depend on an impaired autophagosomes biogenesis, the members of the Ulk-1 complex, namely Ulk1, Fip200, Atg13 and Atg101 were analyzed in whole cell lysates by denaturing and reducing Wb. Furthermore, two phospho-specific antibodies were used to immunodetect phosphoUlk1-Ser555 (pUlk1-Ser555) and phosphoUlk1-Ser757 (pUlk1-Ser757), which represent two phosphorylation sites generally associated with activation and repression of kinase activity and of autophagy induction, respectively.

Fip200 was unaltered among the experimental groups (Figure 5). On the other hand, Atg13, which did not change in Dexa-2d and Dexa-6d cells with respect to Vehicle cells, was instead markedly increased in Dexa-6d* (*p* < 0.0001). Conversely, Ulk1 and Atg101 showed an opposite trend, since they were significantly decreased with respect to Vehicle cells in Dexa-2d and Dexa-6d cells and almost repressed in Dexa-6d* cells. However, in Dexa-6d* cells, Ulk1 turned out to be efficiently phosphorylated at Ser555 (Figure 5); therefore, the pUlk1(Ser555)/Ulk1 ratio was markedly increased in Dexa-6d* cells with respect to vehicle cells (*p* < 0.0001) (Figure 5). Noteworthy, the phosphorylation of Ser 555 results as a specific process, since the pUIk1(Ser757)/Ulk1 ratio was unaltered under all different experimental conditions; only in Dexa-2d and Dexa-6d cells a barely significant increase of this ratio was observed.

The decrease in Ulk-1 and Atg101 and the increase in Atg13 in Dexa-6d* cells raised the question of whether this feature was dependent on transcriptional regulation or on proteolytic turnover. In order to solve this aspect, a RT-PCR analysis of vehicle and Dexa-6d* cells was undertaken to verify the gene expression of Atg101, Atg13, Ulk1 and Beclin1 (Table 1). The analysis ruled out a meaningful variation in the transcription rate for all tested genes, even though Ulk-1, Atg13 and Beclin1 displayed a slight tendency toward an increase (Figure 6).

In order to verify whether the loss of Atg101 and Ulk-1 and the increase in Atg13 in Dexa-6d* cells were determined by an altered protein turnover through the UPS, vehicle and Dexa-6d* cells were stimulated with 300 nM epoxomicin (i.e., a powerful inhibitor of the proteasome chymotrypsin-like activity) over the last four hours of dexamethasone treatment (Figure 7).

A first important observation was that epoxomicin raised the Atg101 and Ulk-1 content in vehicle cells, suggesting that the two proteins are proteasome substrates. This was furtherly supported by the evidence that epoxomicin delivery quickly rescued Atg101 and Ulk-1 content in Dexa-6d* cells, while no effect was observed for Atg13 (Figure 7). In particular, the epoxomicin addition brought the level of Atg101 and Ulk1 back to that of vehicle cells in the absence of epoxomicin (Figure 7). Conversely, epoxomicin only slightly increased the Atg13 level in vehicle cells, but no further increase in this protein was detected in Dexa-treated cells. As a whole, these data suggest a role of UPS in controlling the levels of Atg101 and Ulk-1 but not of Atg13.

In order to verify whether the overall decrease in poly-ubiquitinated proteins, documented in Dexa-6d* cells, was further rescued by epoxomicin, filters were stained with the anti-Ub antibody (Figure 8). In this case, epoxomicin treatment of Dexa-6d* cells brought the content of poly-Ub proteins back to that of vehicle cells treated with epoxomicin, which thus show a similar level of poly-UB protein in the presence of epoxomicin. As a matter of fact, whilst vehicle cells experienced a 2-fold increase in poly-Ub proteins level in the presence of the proteasome inhibitor, Dexa-6d* cells had a 3-fold increase (*p* < 0.0001) (Figure 8).

To study the behavior of a further natural substrate of proteasome substrate, not obviously intertwined with autophagosome biogenesis, IkBα immunostaining in the presence of epoxomicin was assayed in Dexa-6d* cells (Figure 8). Interestingly, IkBα clearance was markedly inhibited in Dexa-6d* cells, and its immunostaining turned out to be more marked than in vehicle cells, suggesting an accumulation of IkBα in Dexa-6d* cells. Accordingly, delivery of epoxomicin stimulated an increase in IkBα in vehicle cells, but not in Dexa-6d* cells (Figure 8).

## 3. Discussion

Stimulation of Trabecular Meshwork Cells by dexamethasone (a prototypical GC) is among the most studied experimental models of glaucoma in vitro. This research tool mirrors the existence of a severe acute clinical form of glaucoma, sustained by a progressive and relevant IOP increase, diagnosed in a subset of subjects who adhere to a GCs-based therapy for pre-existing pathologies [2,3,11,12]. GCs are supposed to induce a dysregulation of TMCs metabolism and a pathological remodeling of TM microarchitecture, which offers resistance to the drainage of aqueous humor. This would cause IOP increase followed by optic nerve degeneration and RGCs apoptosis through mechanical stress [2,7,8,44,45,46].

The rationale behind the scheme and dosage of Dexa administration adopted in this study (with particular reference to the 6-day stimulation with daily 0.1 µM Dexa, i.e., Dexa-6d* cells) relies upon the clinical evidence that subjects develop increased IOP after being treated daily for at least 1 week with GC, often at high doses [12].

First of all, it is worth commenting that the Dexa treatment appears to promote the expression of p21 and p53, two key proteins in regulating cell proliferation and viability [47], something already observed in rat hepatoma cells [48].

The upregulation of p21 at day 2 is likely to contribute to the quiescent and senescent metabolic state progressively acquired by TMCs after GCs exposure, as reported by other authors [46]. Conversely, p53 upregulation on the same day of treatment does not appear to be primarily involved in inducing apoptosis, since phosphorylation at serine 46 does not parallel the total protein increase. Conversely, this phospho-site is likely involved in triggering the apoptotic program in Dexa-6d*, when the phospho-p53/p53 ratio increases with the simultaneous building up of the p25 fragment of PARP (see Figure 1).

However, the expression of p21 and p53 in this TMCs cell model displays some unclear issues; in particular, the drop in p21 after 6 days of Dexa stimulation in the presence of elevated levels of p53, which would be expected to promote instead the expression of p21, demands further studies that fall beyond the original scope of this work.

In this study, we have focused our attention on UPS and autophagy, which are the two proteolytic pathways primarily regulating the homeostasis of eukaryotic cells and, in particular, post-mitotic cells [17,49,50,51,52].

Nonetheless, a progressive loss of proteasome functionality with ageing has been reported in glaucomatous TMCs and this is a very relevant phenomenon which is expected to contribute to TMC dysfunction and glaucoma onset in elder patients.

However, with the exception of a modest decrease in capped assemblies/free 20S ratio in the case of a prolonged stimulation (i.e., Dexa-6d* cells), proteasome composition and bulk-proteolytic activity of TMCs appeared to be substantially unaffected by Dexa treatment, at least under these experimental conditions in vitro [16,25,53,54,55].

Conversely, recent research-based and genetic evidence has already suggested that autophagy might play a pivotal role in RGC neurodegeneration and glaucoma pathogenesis [56]. A genetic linkage indeed exists between mutation in the optineurin gene (OPTN) and RGCs loss. The biological function of OPTN is still largely unclear, but the protein certainly plays a role in autophagy regulation and delivery of substrates to the autophagosomes as well as in the intracellular trafficking of vesicles [5,6,57].

Furthermore, TMCs were reported to modulate autophagy in response to several stimuli ranging from oxidative stress to mechanical stretch [58,59,60]. Main evidence for this is: (i) a decrease in autophagy markers, such as LC3B-II, in TMCs isolated from glaucomatous subjects along with a reduced lysosome basification [18,59]; (ii) a dysregulation of autophagy upon chronic oxidative stress exposure in cultivated TMCs [59]; (iii) the role of autophagy in TGF-β-mediated TM stiffness [61]; (iv) autophagy is a central pathway in retina neurodegeneration during ageing [62]; (v) a defective autophagy flux in the DBA/J2 mouse, a glaucoma murine model characterized by a spontaneous development of IOP [63]; (vi) the protective effect of rapamycin, an autophagy inducer, against RGCs apoptosis and IOP elevation in a rat model of glaucoma [64]. Very recently, a dysregulation of autophagy, based on a reduced content of LC3B-II and Beclin-1, has been reported in TMCs of DBA/J2 mouse treated with GCs and, further, a protective role of rapamycin, again, in TMCs homeostasis and TM remodeling, was uncovered [65,66].

Additionally, TMCs from glaucoma subjects were reported to express high levels of markers of ER stress and of the Unfolded Protein Response (UPR) which is a major metabolic threat [67,68]. Accumulation of unfolded proteins in the endoluminal space of the ER can be handled by the UPS and autophagy in an attempt to restore protein homeostasis, but can also originate from a primary alteration of these pathways.

Accordingly, data herein reported identify a severe dysregulation of autophagy in human primary TMCs stimulated with Dexa. Specifically, LC3B-II building up in the presence of CQ was decreased in Dexa-2d cells with respect to vehicle cells and this drop was even greater in Dexa-6d cells, suggesting that a prolonged exposure to the GC turned into a worsening of the autophagy flux. This possibility was further strengthened by the evidence that when cells were repeatedly exposed to fresh Dexa, as in the case of Dexa-6d* cells, LC3B-II flux towards lysosome was almost abolished (Figure 4).

Transcriptional and molecular investigations allowed us to identify a severe imbalance of the Ulk-1 complex, which is the master regulator of autophagosome biogenesis. Specifically, Ulk-1 kinase and Atg101 bioavailability were progressively downregulated by Dexa, reaching a peak in Dexa-6d* cells.

Conversely, Fip200 and Atg13, which together with Ulk-1 and Atg101 form the whole Ulk-1 complex, were not downregulated in Dexa-2d and Dexa-6d cells; Atg13 level was instead significantly increased in Dexa-6d* cells [32,34,36]. In this framework, although speculative, the increased rate of Ulk1 phosphorylation at Ser555 (probably by AMPK), which usually brings about the activation of the complex, mirrored by an unaltered rate of phosphorylation at Ser757 (likely by mTOR), which usually inhibits kinase activity, would envisage that autophagosome dysfunction in Dexa-6d* cells is not linked to a negative modulation of the Ulk1 complex by upstream signaling pathways [27,34,36,69].

In order to tentatively interpret the findings herein reported, it is worth recalling that Ulk-1 KO mice displayed a very modest phenotypic alteration, whilst Atg13 and Fip200 KO mice displayed lethality already in utero [70,71,72,73].

Furthermore, an Atg13 mutant, which lacks the Ulk1/2 binding site, was able to partially restore autophagy in Atg13-deficient (KO) cells, and structural and molecular studies highlighted that Atg-101-Atg13 forms a heterodimeric complex which triggers autophagosome biogenesis [73,74,75,76,77,78,79]. In this framework, Atg101 is expected to stabilize Atg13 and Ulk-1, protecting them from proteasome-mediated degradation.

Thus, there is compelling evidence that the Atg101-Atg13-Fip200 heterotrimer is more essential than Ulk-1 in activating autophagosome biogenesis, coming into contact with downstream factors and, in particular, with the PtdIns3k complex (made up of Vps24, Beclin-1, Atg15 and Atg14) [30,35,36,75,76,77,80,81,82].

In this scenario, the severe loss of Atg101 observed in Dexa-treated TMCs could be identified as the main cause of the autophagy downregulation of Dexa-treated cells.

The gene expression analysis and the assay in the presence of epoxomicin (i.e., a high affinity and specific proteasome inhibitor) allowed us to attribute the Ulk-1 and Atg101 loss in Dexa-6d* cells to a faster clearance of these proteins by the proteasome. In addition, the overall marked decrease in poly-ubiquitinated proteins immunostaining in all Dexa-treated cells, and in particular, in Dexa-6d* cells, along with the quick rescue of their immunostaining in the presence of epoxomicin, strongly suggesting that the overall rate of protein turnover through the UPS is stimulated by Dexa.

In conclusion, the present study provides some preliminary clues for interpreting the autophagy dysfunction in TMCs challenged with GCs. Data from Dexa-6d and, most notably, from Dexa-6d* cells, which differ for the fact that this last experimental condition received daily fresh doses of the drug, envisage that GCs exposure, especially if repeated over time, progressively reduces the bioavailability of critical intracellular factors for autophagosomes biogenesis (such as Ulk-1 and Atg101).

Since this loss is readily blocked by delivering a proteasome inhibitor, and ubiquitinylation is generally considered as the rate-limiting step of substrates turnover through the UPS, we may propose, as a very preliminary working hypothesis, that these factors could be one or more E3-ligases. Interestingly, modulation of E3-ligase by GCs had been already described in the catabolism of muscle cells through overexpression of MURF [10,83,84,85].

Although this study is limited by having been conducted only on one TMC strain and further approaches are urgently demanded to clarify whether the working hypothesis herein formulated has a general significance in steroid-induced glaucoma pathogenesis, it appears clear that, given the central role played by autophagy in cell metabolism and, more competently than the UPS, in clearing out misfolded proteins and intracellular aggregates, a dysregulation of this pathway is expected to be particularly harmful for TMCs when exposed to noxious stimuli such as GCs.

## 4. Materials and Methods

### 4.1. Cell Culture

The TMC cell line was purchased from Cell Application (San Diego, CA, USA) and grown in complete DMEM high glucose supplemented with 10% FBS plus supplement (antibiotics and non-essential amino acids) (Sigma-Aldrich, St. Louis, Co, USA). All the experimental procedures were carried out within the 9th passage, before the cells acquired a more senescent phenotype (at 12th passage).

Dexamethasone (Dexa), epoxomicin and chloroquine (CQ) were purchased from Sigma-Aldrich (St. Louis Co, USA) and resuspended in pure ethanol (Dexa), DMSO (epoxomicin) or in sterile phosphate buffered saline (PBS) before use (CQ).

Notably, the whole experiments reported herein as Dexa-6d* were also performed by delivering 500 ng/mL (1 µM Dexa) every two days for 6 days. However, experimental outcomes were fully comparable to those obtained in the presence of 0.1 µM Dexa. Thus, only data coming from 0.1 µM Dexa will be discussed.

Furthermore, preliminary experimental settings clarified that experimental outcomes among vehicle-treated cells and between vehicle cells and untreated cells were fully comparable. Hence, for the sake of clarity, unless otherwise indicated, the results from only one vehicle group (hereafter referred as vehicle) are presented in the following discussion and in the results’ figures.

In all experimental conditions, TMCs were not allowed to reach confluency. To avoid this possibility, in some experimental settings, cells were trypsin-detached and split into new flasks without affecting the overall experimental outcome.

### 4.2. MTT Assay

For 3-(4,5-Dimethylthiazol-2-Yl)-2,5-Diphenyltetrazolium Bromide (MTT)assay (Sigma-Aldrich, St. Louis, Co, USA), cells were seeded into a Costar 96-well plate. Stimuli were delivered as indicated above. At specific timepoints, MTT was dissolved (5 mg/mL) in sterile PBS and administered to cells. After 2h incubation at 37 °C, the extraction buffer was added (N,N-dimethyl-formamide, diluted 1:1 with sterile deionized water, taken up at pH 4.7 with a 80% acetic acid and 2.5% 1N HCl solution) and optical densities (O.D.) were read at 470 nm through an infinite M200 Tecan spectrophotometer.

### 4.3. Immuno-Fluorescence Microscopy

TMCs were seeded on IF cover slips and grown at 37 °C, 5% CO_2_ in the absence or presence of Dexa as indicated above. At the end of the treatment, cells were washed in PBS and fixed in 4% paraformaldehyde for 10 min at room temperature (R.T.). Then, the cells were washed twice with PBS and incubated in PBS + 0.03% Triton + 3% BSA for 30 min at R.T. Thereafter, cells were washed twice with PBS and incubated overnight at 4 °C with an anti-LC3B primary antibody. The following day, after two washings with PBS of 15 min each at R.T., cells were incubated with Alexa Fluor-conjugated specific secondary antibodies for 1 h at R.T. Finally, after two washings with PBS, each for 15 min at R.T., the coverslips were mounted and the images were captured through a Zeiss Axioplan 2 fluorescence microscope connected to a digital camera. The number of LC3B-positive dots was quantified by using the ImageJ plug-in based on the Watershed algorithm.

### 4.4. Proteasome Assay and Native Gel Electrophoresis

Crude cell extracts (e.g., soluble fraction of the cell) were extracted under non-denaturing conditions through freeze–thawing cycles in 250 mM sucrose, 20% glycerol, 25 mM Tris-HCl, 5 mM MgCl_2,_ 1 mM EDTA, 1 mM DTT, 2 mM ATP, and pH 7.4, as described elsewhere [40,86]. Thereafter, lysates were cleared by centrifugation at 13,000 rpm, for 20 min, at 4 °C and protein concentration was normalized by Bradford assay.

In the case of the proteasome assay, 20 µg of proteins were diluted in 20% glycerol, 25 mM Tris-HCl, 5 mM MgCl_2,_ 1 mM EDTA, 1 mM DTT, 2 mM ATP, and pH 7.4 in the presence or absence of 500 nM epoxomicin in a Corning 96-well Black Microplate. Reaction mixtures were pre-incubated for 30 min at 37 °C. Thereafter, 50 µM 7-amino-4-methylcoumarin (AMC) labeled Suc-Leu-Leu-Val-Tyr-AMC peptide (referred to as LLVY-amc) (Boston Biochem, Boston, MA, USA) was delivered to each well and the release of fluorescence was monitored over 2h (in any case, until linearity was observed). Obtained values, expressed as nmol substrate/min, were calculated and plotted. The rate of peptide hydrolysis (negligible) in the presence of epoxomicin was subtracted from that in the absence of the proteasome inhibitor. Each experimental condition was run in triplicate in the same plate. The slopes of each curve were then plotted and compared at each timepoint.

In the case of native gel electrophoresis, 75 µg of proteins from each experimental condition were separated through a 3.5% acrylamide gel under native conditions. Gels were then harvested and soaked in a clean dish in the reaction buffer (50 mM Tris, 5 mM MgCl_2_, 1 mM ATP, pH 7.5) supplemented with 75 µM LLVY-amc.

Proteins were then transferred to a HyBond-ECL nitrocellulose filters (see also below for details) and probed with an antibody specific proteasome subunits α7 and Rpn10 (Protein-tech Group, Manchester, UK), diluted 1:3000 in 0.02% Tween-PBS fat-free milk and, after, incubated with a horseradish peroxidase-conjugated anti-rabbit or anti-mouse IgG antibody (Bio-Rad, Hercules, CA, USA), diluted 1:50,000 in 0.2% Tween-PBS fat-free milk.

### 4.5. Western Blotting

For denaturing and reducing Wb, cell pellets were lysed in RIPA buffer and cleared by centrifugation at 13,000 rpm for 30 min, at 4 °C. Protein concentration was normalized by the Bradford assay. Depending on the target, 5 up to 40 µg of proteins per lane were loaded.

Protein transfer to filters was carried out as described in the previous paragraph. The various antibodies used were administered following the manufacturer’s indication and developed as described in the previous section. Antibodies for autophagy-related proteins as well as anti-Ub antibody were all purchased from Cell Signaling Technologies. Myocilin antibody was purchased from Sigma-Aldrich (clone 4F8) (St. Louis, Co, USA).

Inhibitor of kappa B (IkBα) antibody was purchased from Santa Cruz Technologies. Cleaved Poly-ADP ribose polymerase (PARP) p25 fragment antibody together with p53 and a phospho-specific antibody for p53 phosphorylated at serine 46 ((phospho-p53(Ser46)) were purchased from Abcam, Oxford, UK.

Remarkably, preliminary experimental settings revealed that, in some cases, the micrograms of proteins to be loaded to obtain a valid signal (e.g., >30 µg) were off the linearity range of antibodies raised against internal controls, e.g., β-actin and GAPDH (Protein-tech Group, Manchester, UK) (see Appendix A). Therefore, fold-change in individual proteins was calculated by normalizing the given protein to the total proteins loaded in each lane. To this aim, filters were stained with Ponceau-Red S and gels by Coomassie Brilliant Blau (CBB) (this last stain is not shown).

### 4.6. Gene Expression Analysis

RNA from every experimental condition was isolated with Trizol reagent (Life-Technologies). First strand cDNAs were synthesized from 1 µg of total RNA in a 20 µL reaction with reverse transcriptase according to the manufacturer’s instructions (BioLine). Thereafter, real-time PCR was performed on 30 ng of cDNA, using a SYBR green Master Mix (Bio-Rad). GAPDH was used as the internal control. All primers used in these experiments are reported in Table 1. In the case of the Atg101 gene (C12orf44), primers were purchased from Bio-Rad Laboratories. The relative transcription mRNA level was calculated for each gene by using the 2-ΔΔCt formula and data are reported as fold-increase with respect to the control group, as indicated in Figure 6.

### 4.7. Statistical Analysis

In all cases, the values reported are expressed as mean ± SD. With the exception of the gene expression analysis (see Figure 6), which is expressed as mean ± SEM. An unpaired τ Student’s test, non-parametric Mann–Whitney one-way ANOVA followed by Tukey’s post hoc significance test were used. Statistical significance was attributed to differences characterized by *p* < 0.05. In the absence of a capped lane, asterisks of statistics refer to the comparison between that given experimental observation vs. the corresponding vehicle-treated condition. Data elaboration and statistical analysis were performed by using the GraphPad Prism software (6.0 version).

## Figures and Tables

**Figure 1 ijms-22-05891-f001:**
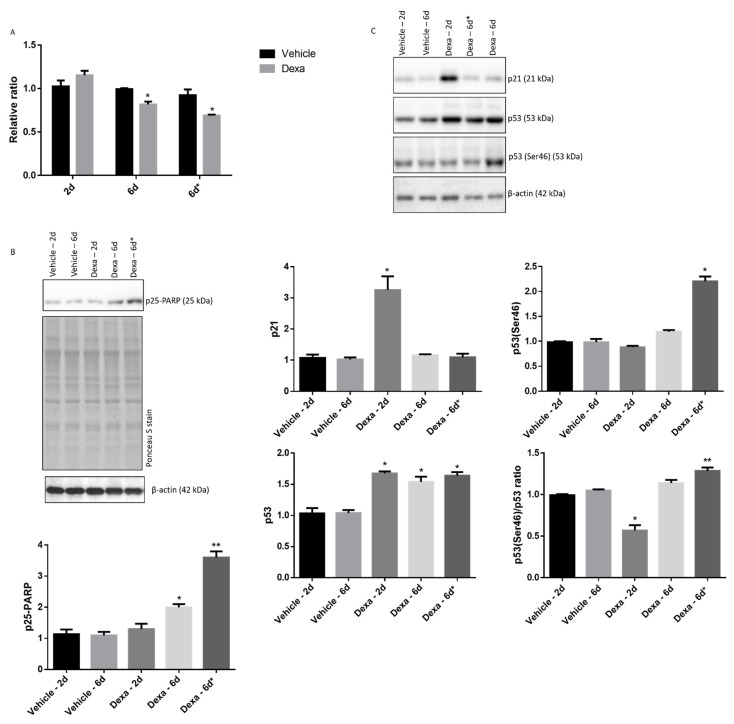
Reduced growth and apoptosis in TMCs repeatedly stimulated with Dexa. (**A**) TMCs were seeded at the same dilution (3 wells for each experimental condition) and challenged with vehicle or Dexa following the scheme and dosage indicated. At selected timepoints, cell proliferation and viability were assessed by MTT assay. A nominal value of 1 was assigned to the O.D. of the first vehicle well. A representative experiment of two independent observations is reported (*n* = 3). Comparisons have been run between Dexa and correspondent vehicle cells at each timepoint. * *p* < 0.005, unpaired τ Student’s test. (**B**) Immunodetection of the p25 fragment of PARP in whole cell lysates by Wb. Although β-actin pattern is shown, normalization was performed on total proteins (Ponceau S stain). A representative experiment of three independent observations is reported (*n* = 3) * *p* < 0.04, ** *p* < 0.0008. (**C**) Immunodetection of the p21, p53, phospho-p53(Ser46) and determination of the phospho-p53(ser46)/p53 ratio in whole cell lysates by Wb. Normalization was performed on β-actin. A representative experiment of three independent observations is reported (*n* = 3) * *p* < 0.0001; ** *p* < 0.008; one-way ANOVA followed by Tukey’s post hoc significance test.

**Figure 2 ijms-22-05891-f002:**
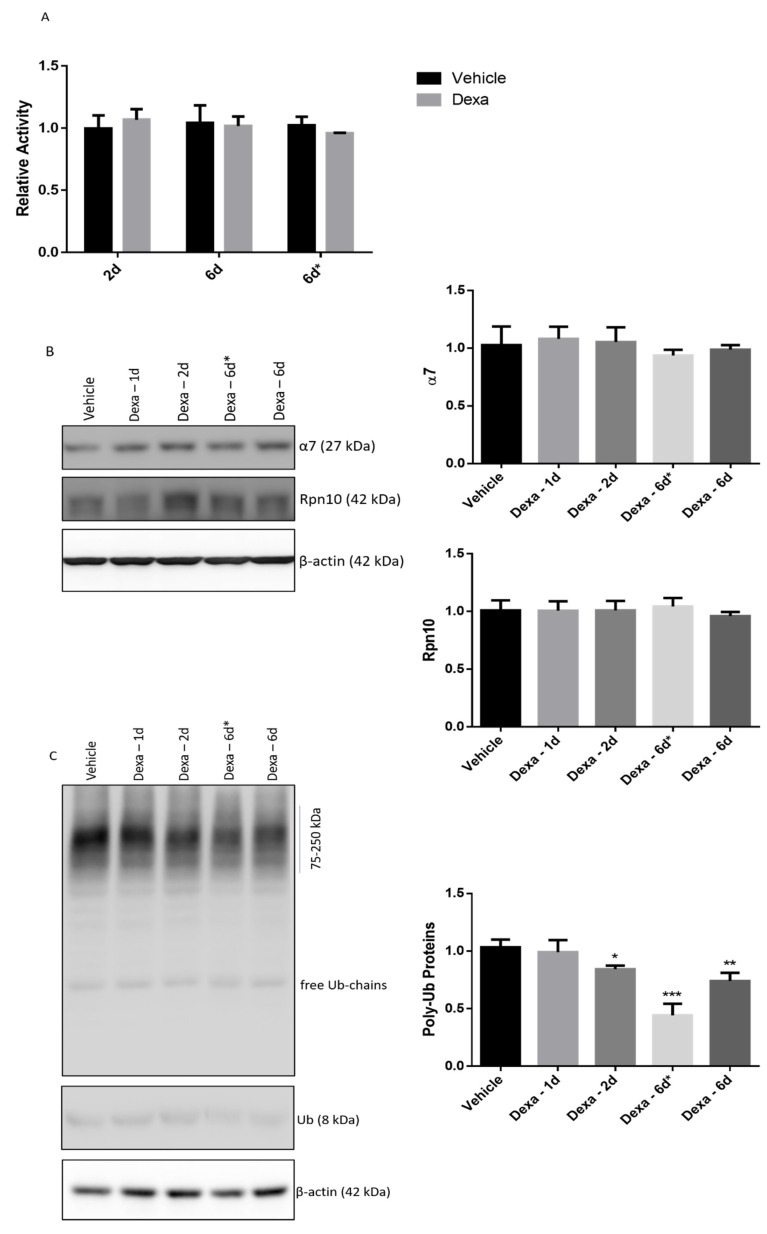
Proteasome activity is not significantly altered by Dexa treatment. (**A**) Proteasome assemblies of crude cell extracts of TMCs were assayed for the kinetics of LLVY-amc cleavage. The release of fluorescence (λ_exc._: 380 nm; λ_em._: 460 nm) was monitored until linearity was observed (typically 2 h). Individual slopes were calculated by subtracting the fluorescence release in the presence of 500 nM epoxomicin (null). Data are presented as fold variation of the slope between Dexa and correspondent vehicle-treated cells. A nominal value of 1 was assigned to the slope calculated for the first (1 out of 3 wells) vehicle cells. Comparisons have been run between Dexa and correspondent vehicle cells at each timepoint. (**B**) Immunodetection of α7 and Rpn10 subunits and (**C**) ubiquitinated proteins by denaturing and reducing Wb in whole cell lysates. β-actin was used as internal control. Notably, (**C**) has been manipulated to clear out one lane loaded with an additional vehicle cell lysate. * *p* < 0.04; ** *p* < 0.005; *** *p* < 0.0001. A representative experiment of three independent observations is reported (*n* = 3); one-way ANOVA followed by Tukey’s post hoc significance test.

**Figure 3 ijms-22-05891-f003:**
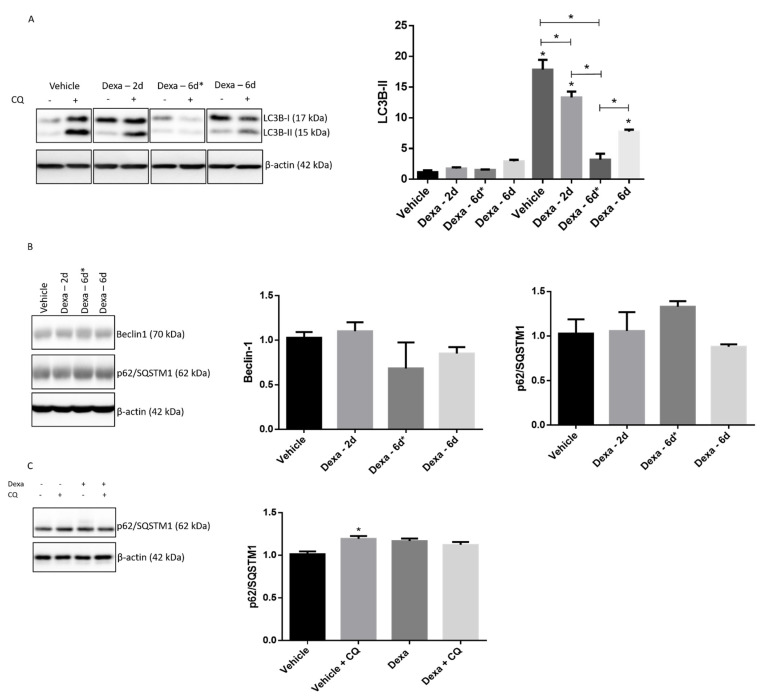
Impaired autophagy flux in Dexa-treated TMCs. Immunoblotting of LC3B in the presence/absence of 20 µM CQ (**A**) and of Beclin-1 and p62/SQSTM1 (**B**) in whole cell lysates of TMCs * *p* < 0.0001. (**C**) p62/SQSTM1 accumulation in vehicle-treated cells and Dexa-6d* cells in the presence of 20 µM CQ. Normalization in both figures was performed on β-actin. * *p* < 0.005. A representative experiment of three independent observations is reported (*n* = 3); one-way ANOVA followed by Tukey’s post hoc significance test.

**Figure 4 ijms-22-05891-f004:**
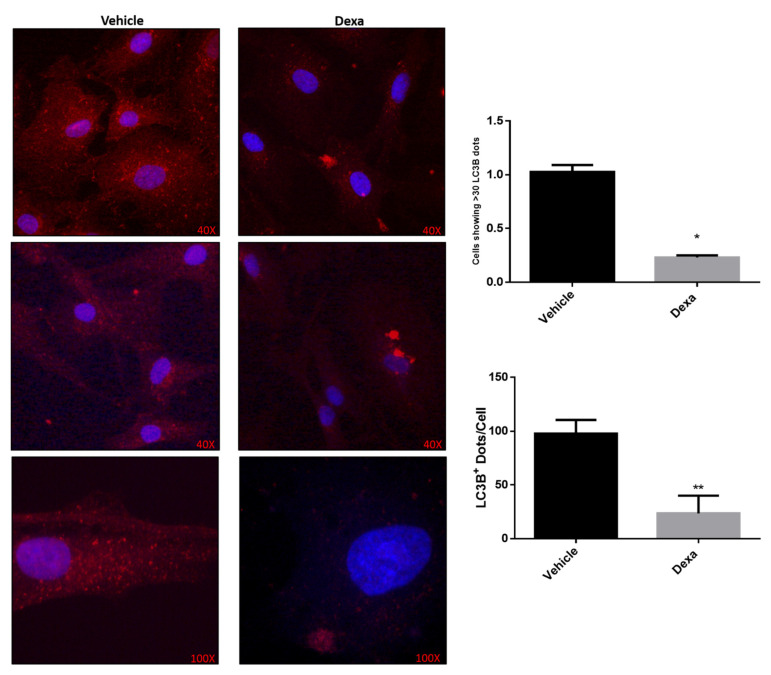
Decreased autophagosomes count in Dexa-treated TMCs. (A) Immunofluorescence microscopy analysis of LC3B^+^ autophagosomes (e.g., red spheres) in vehicle and Dexa-6d* cells. Nuclei were stained by Hoechst. Images were acquired at 40X and 100X magnification. Autophagosome count was analyzed by means of either percentage of cells showing at least 10 LC3B^+^ dots and average number of dots per cell. * *p* < 0.0001; ** *p* < 0.004. A representative experiment of two independent observations is reported (*n* = 10); unpaired Students’ τ test.

**Figure 5 ijms-22-05891-f005:**
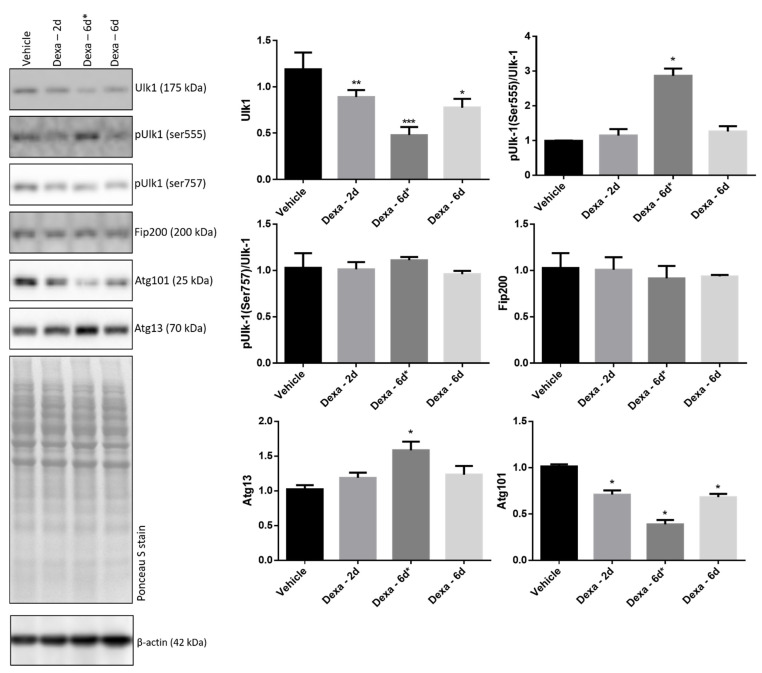
Downregulation of ULK1 members in Dexa-treated TMCs. Immunoblotting of Ulk-1, pUlk1(Ser555), pUlk1(Ser757), Fip200, Atg13, and Atg101 in whole cell lysates of TMCs. Normalization was performed on total proteins (Ponceau S stain). * *p* < 0.0001, ** *p* < 0.009, *** *p* < 0.001. A representative experiment of three independent observations is reported (*n* = 3); one-way ANOVA followed by Tukey’s post hoc significance test.

**Figure 6 ijms-22-05891-f006:**
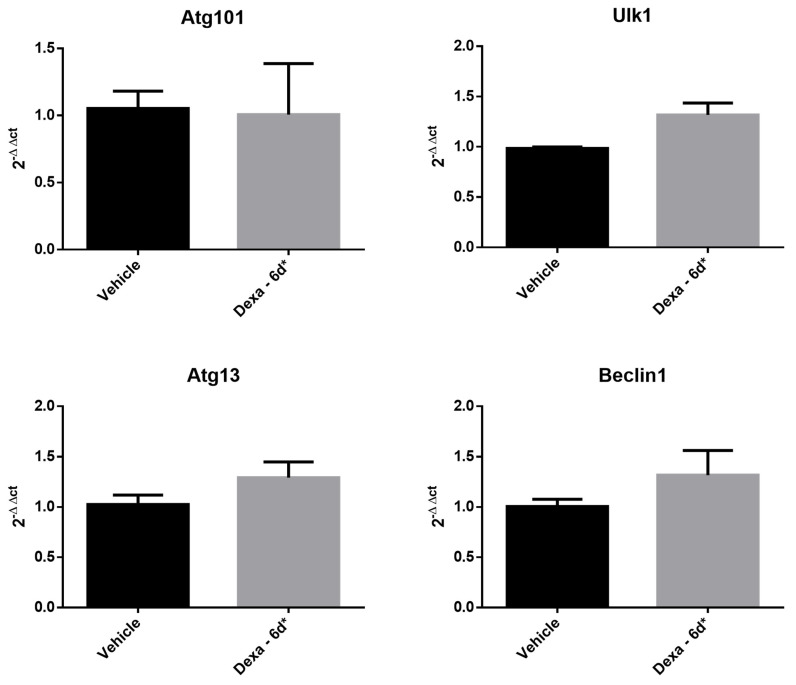
Autophagy gene expression analysis of Dexa-treated TMCs. RT-PCR results of the expression of ULK-1, ATG101, ATG13, and Beclin-1. Gapdh was used as internal control, as indicated in the Section 4. Data are reported as fold-increase with respect to control group. The 2^−ΔΔCt^ formula was used to calculate the fold-increase versus vehicle cells.

**Figure 7 ijms-22-05891-f007:**
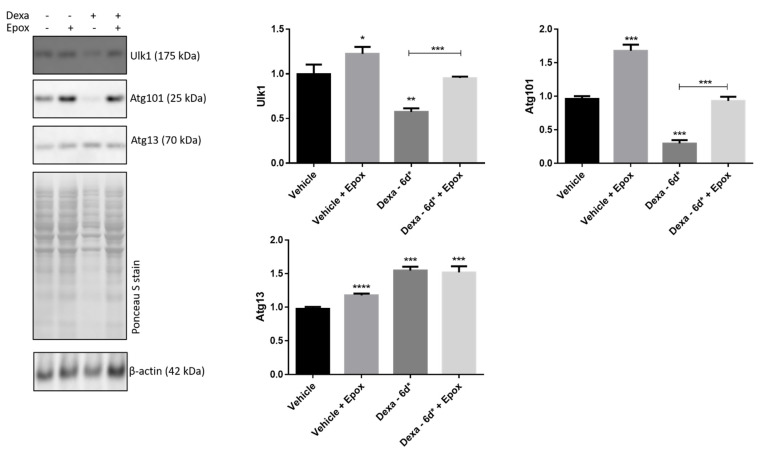
Enhanced turnover of Ulk-1 and Atg101 in Dexa-treated TMCs. (A) Immunoblotting of Ulk-1, Atg101 and Atg13 in whole cell lysates of TMCs. Normalization was performed on total proteins (Ponceau S stain). * *p* < 0.02; ** *p* < 0.0004; *** *p* < 0.0001; **** *p* < 0.001. A representative experiment of three independent observations is reported (*n* = 3); one-way ANOVA followed by Tukey’s post hoc significance test.

**Figure 8 ijms-22-05891-f008:**
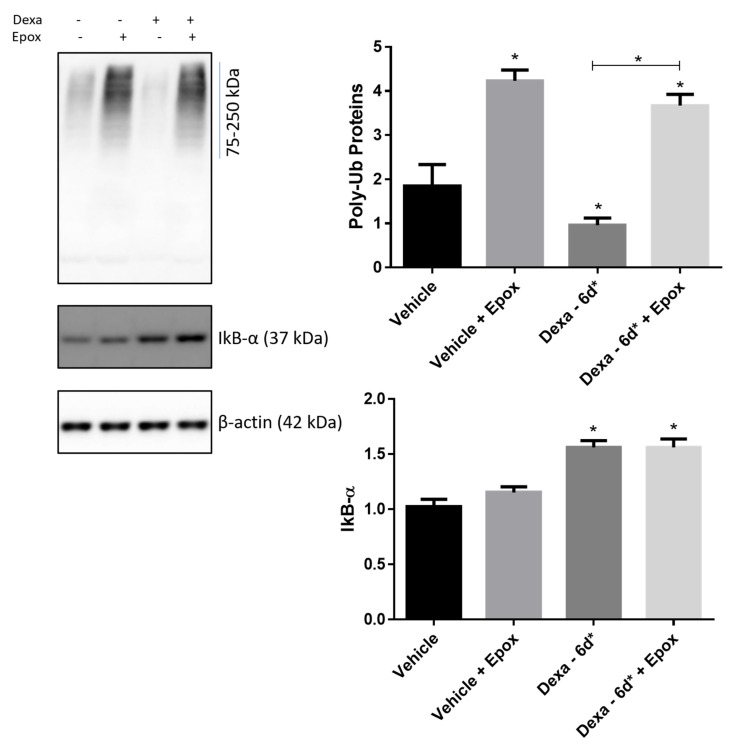
Turnover of poly-Ub protein and IkBα in Dexa-treated TMCs. Immunoblotting of Ub and IkBα in whole cell lysates of TMCs. β-actin was used as internal control. A representative experiment of three independent observations is reported (*n* = 3); * *p* < 0.0001; one-way ANOVA followed by Tukey’s post hoc significance test.

**Table 1 ijms-22-05891-t001:** Human-specific primer sequence used in RT-PCR studies.

Gene Name	Sequence
Atg13	Fwd-CCCAGGACAGAAAGGACCTG Rev-AACCAATCTGAACCCGTT
ULK-1	Fwd-GGACACCATCAGGCTCTTCC Rev-GAAGCCGAAGTCAGCGATCT
Beclin-1	Fwd-GGCTGAGAGACTGGATCAGG Rev-CTGCGTCTGGGCATAACG
GAPDH	Fwd-AGAAGGCTGGGGCTCATTTG Rev-AGGGGCCATCCACAGTCTTC

## Data Availability

All data herein reported, including uncropped gel images and GraphPad files, will be available upon request to diego.sbardella@fondazionebietti.it. The primary cell line actually used is stored in cryo-vials and will be available upon request.

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
