# Peer review of "Dexamethasone Downregulates Autophagy through Accelerated Turn-Over of the Ulk-1 Complex in a Trabecular Meshwork Cells Strain: Insights on Steroid-Induced Glaucoma Pathogenesis"

_ijms, 2021, doi:10.3390/ijms22115891_

Round 1

Reviewer 1 Report

This manuscript investigates the ubiquitin proteasome system (UPS) and autophagy upon dexamethasone treatment in trabecular meshwork cells. The authors conclude that dexamethasone treatment triggers proteasome-mediated degradation of the autophagy initiation ULK1 complex and decreased autophagy formation.

The manuscript is extremely difficult to follow and would benefit extensive editing and shorting out. It lacks a clear logical focus and rationale of the experiments is not properly stated.

There are multiple statements throughout the manuscripts that are incorrect in all of the topics mentioned (glaucoma, UPS and autophagy). Authors indistinctively exchange ocular hypertension and glaucoma. Mechanisms of OHT in the trabecular meshwork cells are/might not be the same as those observed in RGC in glaucoma. OTN is associated to normal tension glaucoma. Author does not mention its role in autophagy.

Author treats induced expression of MYOC as DEX-induced glaucoma. These are different.

There are two manuscripts related to steroid-induced glaucoma and autophagy, which are not discussed. Similarly, the reported role of ER stress in GC-induced glaucoma has not been mentioned.

There is no clear explanation of the logic behind the different Dex-treatment, and the authors have not commented on the different results obtained with the different treatments.

Figure 1, 2, 3 and text associated are extremely difficult to follow. Proteasome activity can be easily monitored and summarized in one figure.

Results in Figure 4 (LC3-II and CQ) were not properly interpreted. As shown, results do not show decrease flux. P62 should be also be evaluated in the presence/absence of CQ.

The decrease in LC3 observed by IF in Dex-treated cells (Fig 5) is not confirmed in Figure 4

Ponceau stain is not sensitive enough for normalization studies. It must be substituted by non-saturated actin or GAPDH blots.

Results do not support the conclusions. Does epox treatment rescue phenotype?

Discussion is extremely speculative.

MW must be added to the blots

There is no information on how many independent cell strains were used. As per the data availability statement just one cell line was used. All the experiments must be repeated with at least three independent cell lines.

As per Material & Methods experimental procedures were conducted with cells at passage 12th. Primary cultures of human TM cells at that passage are senescence.

Methods Section should strictly describe methods used in the manuscript (i.e. dot blots were described, but no dot blot was included). All discussion or background information must be deleted from Methods

Reviewer 2 Report

This is a well designed and technically well conducted study that provides important information on the molecular mechanism responsible for the pathogenesis of steroid-induced glaucoma, specifically regarding the role played by alterations of the two main mechanisms (UPS and autophagy) that collaborate in carrying out the controlled degradation of cellular proteins. I therefore recommend the publication of the manuscript with only minor revisions:
1- In supplementary figure 1, the third bar of graphs A and B refers to the 6 day vehicle (and not as indicated to Dexa 6-d *). Furthermore, in the text the supplementary figure 1B is erroneously indicated as supplementary figure 2 (line 162).
- In figure 2D the third bar of the graph refers to Dexa 1 micromolar (instead of 0.1 as reported).
- For future studies I suggest to the authors to use a concentration of epoxomicin higher than that used here to completely inhibit in vivo the proteolytic activity of the proteasome (300 nM does not completely inhibit proteasomal degradation especially for substrates whose hydrolysis is also conspicuously operated by the trypsin-like site. See for example Kisselev AF et al., JBC, 2006, 281:8582-90).

Author Response

Reviewer #2

Reviewer: This is a well designed and technically well conducted study that provides important information on the molecular mechanism responsible for the pathogenesis of steroid-induced glaucoma, specifically regarding the role played by alterations of the two main mechanisms (UPS and autophagy) that collaborate in carrying out the controlled degradation of cellular proteins. I therefore recommend the publication of the manuscript with only minor revisions:

Authors: We acknowledge the reviewer for his/her positive revision of our manuscript.

Reviewer: 1- In supplementary figure 1, the third bar of graphs A and B refers to the 6 day vehicle (and not as indicated to Dexa 6-d *). Furthermore, in the text the supplementary figure 1B is erroneously indicated as supplementary figure 2 (line 162).

- In figure 2D the third bar of the graph refers to Dexa 1 micromolar (instead of 0.1 as reported).

Authors: We apologize for these mistakes, we have amended them.

Reviewer: - For future studies I suggest to the authors to use a concentration of epoxomicin higher than that used here to completely inhibit in vivo the proteolytic activity of the proteasome (300 nM does not completely inhibit proteasomal degradation especially for substrates whose hydrolysis is also conspicuously operated by the trypsin-like site. See for example Kisselev AF et al., JBC, 2006, 281:8582-90).

Authors: This is very precious advise, we absolutely agree with the reviewer’s position about this topic. We originally did the experiments also at 0.5 µM and 1 µM epoxomicin, and we could not see significant qualitative or quantitative variation of experimental results regarding Atg101 or Ulk-1 rescue. However, at these higher Epoxomicin concentrations, but not at 300 nM, we actually observed a paradoxical NF-kB activation (namely an increased phosphorylation of IkBα at Serine 32 and 36), a finding already reported by other authors (Lee KH, Jeong J, Yoo CG. Long-term incubation with proteasome inhibitors (PIs) induces IκBα degradation via the lysosomal pathway in an IκB kinase (IKK)-dependent and IKK-independent manner. J Biol Chem. 2013 Nov 8;288(45):32777-32786. doi: 10.1074/jbc.M113.480921. Epub 2013 Oct 1. PMID: 24085292). Since IkBα was analysed in this study, this epoxomicin concentration looked more adapt to the study.

Reviewer 3 Report

it is a interesting study and well structured

however some figures needs to be improved I know it is not easy but still it should be clear

P value significance is different in every figure and it nay give some misunderstaning to the readers

also some figures have very low quality

example

figure 1 b  are you sure 6d is that much high?  it looks pretty similar 6d* and 6d dont you have a better one?

figure a, 2c  so hard to tell the difference

as i mentioned there are quality problems for some figures it should be improved

Author Response

Reviewer #3

Reviewer: it is a interesting study and well structured

Authors: We are grateful to the reviewer for his/her positive revision of our manuscript.

Reviewer: however some figures needs to be improved I know it is not easy but still it should be clear P value significance is different in every figure and it nay give some misunderstaning to the readers

Authors: If we have meant in the right way the reviewer’s suggestion, we have revised the representation of statistical analysis throughout the text. P values have been approximated and in Materials and Methods we have better explained how significance was determined.

Reviewer: also some figures have very low quality

example

figure 1 b  are you sure 6d is that much high?  it looks pretty similar 6d* and 6d dont you have a better one?

Authors: According to the reviewer suggestion, the figure has been re-edited. We have used a new antibody batch of the original one which did work much better. Clearly the pattern was similar but quantification was made on three observations, being represented as Mean +/- SD (n=3).

Reviewer: figure a, 2c  so hard to tell the difference

Authors: We fully agree that the difference is subtle, but we do not actually claim that Dexa modulates proteasome composition and/or activity. Although small, the difference was reproducible and actually reached statistical significance.

Reviewer 4 Report

Review comments:

  1. Results 2.1. Prolonged TMCs stimulation by Dexa reduces cell growth and induces apoptosis. The authors should provide more evidences and experiments to validate the results. The methodology applied for the evaluation of neuroprotection is insufficient. The authors could include assays to show differential expression of apoptosis-related proteins such as p-p53, p53, p15, p21 and accumulation of damage, among others. Also, the cell viability assay should be assayed by the more quantitative methods such as MTT or WST assay.
  2. The authors should increase the quality of the WB figures in Result 2.2 and the proteasome activity should be validation by other assays such as proteasome activity assay kit and results in quantitative data. In figure D, the x-value should be Dexa (I uM)-1d.
  3. In Result 2.3, the immunofluorescence data showed that the red vesicles were less frequent in Dexa-6d* cells than in Vehicle-cells, but in 40X figure of Dexa group, the intensity showed very high, please represent the consistency figures. The authors should point out the dots and verify the counting methods.
  4. In line 307-309, the Atg13 was up-regulation, and not reducing Wb.
  5. The results and the conclusion are consistent, while the quality of data should be improved.

Author Response

Reviewer 4

Reviewer: Suggestions for Authors

Review comments:

  1. Results 2.1. Prolonged TMCs stimulation by Dexa reduces cell growth and induces apoptosis. The authors should provide more evidences and experiments to validate the results. The methodology applied for the evaluation of neuroprotection is insufficient. The authors could include assays to show differential expression of apoptosis-related proteins such as p-p53, p53, p15, p21 and accumulation of damage, among others. Also, the cell viability assay should be assayed by the more quantitative methods such as MTT or WST assay.

Authors: First of all, we acknowledge the reviewer for his/her positive revision of our manuscript. We have tried to address most of the points raised.

We are particularly grateful for this criticism, since it has allowed us to clarify that the reduced proliferation of Dexa-treated TMCs and the increase of apoptosis rate after prolonged stimulation could, at least in part, rely on p21 and p53 up-regulation. As a matter of fact, p21 immunostaining turned out to increase after 2-days of Dexa treatment, whilst it turned back to basal levels after 6-days. Conversely, p53 immunostaining was persistently increased over treatment, but phosphorylation at serine 46 was increased only in the presence of prolonged Dexa stimulation. It is our personal hypothesis that the pattern of p21 and p53, though interesting, shows some controversies and it is worth being explored at a greater molecular detail in future studies, also considering that we are no expert in this field. We have commented this in the discussion. 

Cell viability was measured again by MTT assay and now included in the results section, whereas the Trypan blue count has been moved to Suppl. Information.

Reviewer:

  1. The authors should increase the quality of the WB figures in Result 2.2 and the proteasome activity should be validation by other assays such as proteasome activity assay kit and results in quantitative data. In figure D, the x-value should be Dexa (I uM)-1d.

Authors: We acknowledge this criticism and we can reply that the bulk proteasome activity for each experimental condition was actually assayed at the beginning of this study and it is now shown in the new Fig. 2A. The reason why we originally preferred to show the native gel approach, though more difficult to follow, was based on some assumptions about substrate preferences of proteasome. As the reviewer sees, we propose that poly-ubiquitinated proteins drop in the absence of a significant modulation of proteasome activity over time. With respect to this, to monitor bulk proteasome activity in crude-cell extracts may also have some drawbacks (e.g., free 20S activation by salts or heat during assay run). Therefore, native-gel was originally preferred to confirm that there was no increase of proteasome capped assemblies activity or abundance (26S/20S ratio) under all tested conditions. On the other side, we agree that this method, greatly ideal to get information on overall activity of individual particles and on capped assemblies/20S ratio, indeed suffers from lower sensibility for measuring bulk proteasome activity. Thus, in the actual version of the manuscript we moved the native-gel assays off into the supplementary information, leaving only a shortened discussion on bulk proteasome activity.

About the quality of figures, we must recall that this is a Western blotting of a native-gel (3-5% acrylamide gel). Proteasome particles MW range from 700 to 3000 kDa and it is sometimes very hard to get high quality images. We apologize for the mistake in Figure D, we have amended it.

Reviewer:

  1. In Result 2.3, the immunofluorescence data showed that the red vesicles were less frequent in Dexa-6d* cells than in Vehicle-cells, but in 40X figure of Dexa group, the intensity showed very high, please represent the consistency figures. The authors should point out the dots and verify the counting methods.

Authors: We acknowledge the reviewer for this criticism. We agree that the intensity of LC3B immunostaining showed high in Dexa cells, but the image was selected because the cells had an appreciable staining. Otherwise, they would have looked exaggerated. In fact, the immunostaining pattern of Dexa cells is also that in the new figure provided (together with the old one) where the LC3B pattern is very faint.

Clearly, the quantification we did was based on calculation from different area and only the final mean value is provided.

To better quantify the loss of autophagosomes in Dexa-treated cells, we have now used the Watershed plug-in of ImageJ. 

Reviewer:

  1. In line 307-309, the Atg13 was up-regulation, and not reducing Wb.

Authors: We have amended the text accordingly

Reviewer:

  1. The results and the conclusion are consistent, while the quality of data should be improved.

Authors: Again, we are grateful to the reviewer and we have tried to do our best to address the raised points.

Round 2

Reviewer 3 Report

better improved

Author Response

On behalf of the other authors, 

I acknowledge the reviewer for his/her criticism and we have tried to address the point raised. Basically, we have significantly edited and shortened the text, in particular Introduction and Discussion sections. Whole manuscript has been further revised by an English-mother tongue. 

Best regards, 

Dr. F. Oddone

Reviewer 4 Report

none

Author Response

On behalf of the other authors of this manuscript,

I acknolwedge the reviewer for his/her positive consideration of our manuscript. 

Best regards, 

Dr. F. Oddone